# Multi-Prediction Deep Boltzmann Machines

**Ian J. Goodfellow,   Mehdi Mirza,   Aaron Courville,   Yoshua Bengio**
Département d'informatique et de recherche opérationnelle
Université de Montréal
Montréal, QC H3C 3J7
{goodfeli,mirzamom,courvila}@iro.umontreal.ca,
Yoshua.Bengio@umontreal.ca

## Abstract

We introduce the *multi-prediction deep Boltzmann machine* (MP-DBM). The MP-DBM can be seen as a single probabilistic model trained to maximize a variational approximation to the generalized pseudolikelihood, or as a family of recurrent nets that share parameters and approximately solve different inference problems. Prior methods of training DBMs either do not perform well on classification tasks or require an initial learning pass that trains the DBM greedily, one layer at a time. The MP-DBM does not require greedy layerwise pretraining, and outperforms the standard DBM at classification, classification with missing inputs, and mean field prediction tasks.[1]

## 1   Introduction

A deep Boltzmann machine (DBM) [18] is a structured probabilistic model consisting of many layers of random variables, most of which are latent. DBMs are well established as generative models and as feature learning algorithms for classifiers.

Exact inference in a DBM is intractable. DBMs are usually used as feature learners, where the mean field expectations of the hidden units are used as input features to a separate classifier, such as an MLP or logistic regression. To some extent, this erodes the utility of the DBM as a probabilistic model–it can generate good samples, and provides good features for deterministic models, but it has not proven especially useful for solving inference problems such as predicting class labels given input features or completing missing input features.

Another drawback to the DBM is the complexity of training it. Typically it is trained in a greedy, layerwise fashion, by training a stack of RBMs. Training each RBM to model samples from the previous RBM's posterior distribution increases a variational lower bound on the likelihood of the DBM, and serves as a good way to initialize the joint model. Training the DBM from a random initialization generally does not work. It can be difficult for practitioners to tell whether a given lower layer RBM is a good starting point to build a larger model.

We propose a new way of training deep Boltzmann machines called *multi-prediction training* (MPT). MPT uses the mean field equations for the DBM to induce recurrent nets that are then trained to solve different inference tasks. The resulting trained MP-DBM model can be viewed either as a single probabilistic model trained with a variational criterion, or as a family of recurrent nets that solve related inference tasks.

We find empirically that the MP-DBM does not require greedy layerwise training, so its performance on the final task can be monitored from the start. This makes it more suitable than the DBM for

practitioners who do not have extensive experience with layerwise pretraining techniques or Markov chains. Anyone with experience minimizing non-convex functions should find MP-DBM training familiar and straightforward. Moreover, we show that inference in the MP-DBM is useful– the MP-DBM does not need an extra classifier built on top of its learned features to obtain good inference accuracy. We show that it outperforms the DBM at solving a variety of inference tasks including classification, classification with missing inputs, and prediction of randomly selected subsets of variables. Specifically, we use the MP-DBM to outperform the classification results reported for the standard DBM by Salakhutdinov and Hinton [18] on both the MNIST handwritten character dataset [14] and the NORB object recognition dataset [13].

## 2   Review of deep Boltzmann machines

Typically, a DBM contains a set of $D$ input features $v$ that are called the *visible units* because they are always observed during both training and evaluation. When a class label is present the DBM typically represents it with a discrete-valued label unit $y$. The unit $y$ is observed (on examples for which it is available) during training, but typically is not available at test time. The DBM also contains several latent variables that are never observed. These *hidden units* are usually organized into $L$ layers $h^{(i)}$ of size $N_i, i \in \{1, \ldots, L\}$, with each unit in a layer conditionally independent of the other units in the layer given the neighboring layers.

The DBM is trained to maximize the mean field lower bound on $\log P(v, y)$. Unfortunately, training the entire model simultaneously does not seem to be feasible. See [8] for an example of a DBM that has failed to learn using the naive training algorithm. Salakhutdinov and Hinton [18] found that for their joint training procedure to work, the DBM must first be initialized by training one layer at a time. After each layer is trained as an RBM, the RBMs can be modified slightly, assembled into a DBM, and the DBM may be trained with PCD [22, 21] and mean field. In order to achieve good classification results, an MLP designed specifically to predict $y$ from $v$ must be trained on top of the DBM model. Simply running mean field inference to predict $y$ given $v$ in the DBM model does not work nearly as well. See figure 1 for a graphical description of the training procedure used by [18].

The standard approach to training a DBM requires training $L + 2$ different models using $L + 2$ different objective functions, and does not yield a single model that excels at answering all queries. Our proposed approach requires training only one model with only one objective function, and the resulting model outperforms previous approaches at answering many kinds of queries (classification, classification with missing inputs, predicting arbitrary subsets of variables given the complementary subset).

## 3   Motivation

There are numerous reasons to prefer a single-model, single-training stage approach to deep Boltzmann machine learning:

1. **Optimization** As a greedy optimization procedure, layerwise training may be suboptimal. Small-scale experimental work has demonstrated this to be the case for deep belief networks [1].

   In general, for layerwise training to be optimal, the training procedure for each layer must take into account the influence that the deeper layers will provide. The layerwise initialization procedure simply does not attempt to be optimal.

   The procedures used by Le Roux and Bengio [12], Arnold and Ollivier [1] make an optimistic assumption that the deeper layers will be able to implement the best possible prior on the current layer's hidden units. This approach is not immediately applicable to Boltzmann machines because it is specified in terms of learning the parameters of $P(h^{(i-1)}|h^{(i)})$ assuming that the parameters of the $P(h^{(i)})$ will be set optimally later. In a DBM the symmetrical nature of the interactions between units means that these two distributions share parameters, so it is not possible to set the parameters of the one distribution, leave them fixed for the remainder of learning, and then set the parameters of the other distribution. Moreover, model architectures incorporating design features such as sparse connections,

pooling, or factored multilinear interactions make it difficult to predict how best to structure one layer's hidden units in order for the next layer to make good use of them.

2. **Probabilistic modeling** Using multiple models and having some models specialized for exactly one task (like predicting $y$ from $v$) loses some of the benefit of probabilistic modeling. If we have one model that excels at all tasks, we can use inference in this model to answer arbitrary queries, perform classification with missing inputs, and so on. The standard DBM training procedure gives this up by training a rich probabilistic model and then using it as just a feature extractor for an MLP.

3. **Simplicity** Needing to implement multiple models and training stages makes the cost of developing software with DBMs greater, and makes using them more cumbersome. Beyond the software engineering considerations, it can be difficult to monitor training and tell what kind of results during layerwise RBM pretraining will correspond to good DBM classification accuracy later. Our joint training procedure allows the user to monitor the model's ability of interest (usually ability to classify $y$ given $v$) from the very start of training.

## 4   Methods

We now described the new methods proposed in this paper, and some pre-existing methods that we compare against.

### 4.1   Multi-prediction Training

Our proposed approach is to directly train the DBM to be good at solving all possible variational inference problems. We call this *multi-prediction training* because the procedure involves training the model to predict any subset of variables given the complement of that subset of variables.

Let $\mathcal{O}$ be a vector containing all variables that are observed during training. For a purely unsupervised learning task, $\mathcal{O}$ is just $v$ itself. In the supervised setting, $\mathcal{O} = [v, y]^T$. Note that $y$ won't be observed at test time, only training time. Let $\mathcal{D}$ be the training set, i.e. a collection of values of $\mathcal{O}$. Let $S$ be a sequence of subsets of the possible indices of $\mathcal{O}$. Let $Q_i$ be the variational (e.g., mean-field) approximation to the joint of $O_{S_i}$ and $h$ given $\mathcal{O}_{-S_i}$.

$$Q_i(\mathcal{O}_{S_i}, h) = \mathrm{argmin}_Q D_{KL} \left( Q(\mathcal{O}_{S_i}, h) \| P(\mathcal{O}_{S_i}, h \mid \mathcal{O}_{-S_i}) \right).$$

In all of the experiments presented in this paper, $Q$ is constrained to be factorial, though one could design model families for which it makes sense to use richer structure in $Q$. Note that there is not an explicit formula for $Q$; $Q$ must be computed by an iterative optimization process. In order to accomplish this minimization, we run the mean field fixed point equations to convergence. Because each fixed point update uses the output of a previous fixed point update as input, this optimization procedure can be viewed as a recurrent neural network. (To simplify implementation, we don't explicitly test for convergence, but run the recurrent net for a pre-specified number of iterations that is chosen to be high enough that the net usually converges)

We train the MP-DBM by using minibatch stochastic gradient descent on the *multi-prediction* (MP) objective function

$$J(\mathcal{D}, \theta) = - \sum_{\mathcal{O} \in \mathcal{D}} \sum_i \log Q_i(\mathcal{O}_{S_i})$$

In other words, the criterion for a single example $\mathcal{O}$ is a sum of several terms, with term $i$ measuring the model's ability to predict (through a variational approximation) a subset of the variables in the training set, $\mathcal{O}_{S_i}$, given the remainder of the observed variables, $\mathcal{O}_{-S_i}$.

During SGD training, we sample minibatches of values of $\mathcal{O}$ and $S_i$. Sampling $\mathcal{O}$ just means drawing an example from the training set. Sampling an $S_i$ uniformly simply requires sampling one bit (1 with probability 0.5) for each variable, to determine whether that variable should be an input to the inference procedure or a prediction target. To compute the gradient, we simply backprop the error derivatives of $J$ through the recurrent net defining $Q$.

See Fig. 2 for a graphical description of this training procedure, and Fig. 3 for an example of the inference procedure run on MNIST digits.

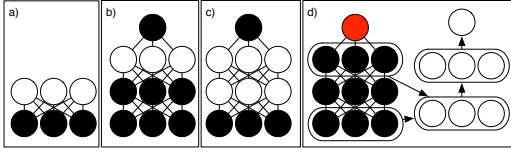

Figure 1: The training procedure used by Salakhutdinov and Hinton [18] on MNIST. a) Train an RBM to maximize $\log P(v)$ using CD. b) Train another RBM to maximize $\log P(h^{(1)}, y)$ where $h^{(1)}$ is drawn from the first RBM's posterior. c) Stitch the two RBMs into one DBM. Train the DBM to maximize $\log P(v, y)$. d) Delete $y$ from the model (don't marginalize it out, just remove the layer from the model). Make an MLP with inputs $v$ and the mean field expectations of $h^{(1)}$ and $h^{(2)}$. Fix the DBM parameters. Initialize the MLP parameters based on the DBM parameters. Train the MLP parameters to predict $y$.

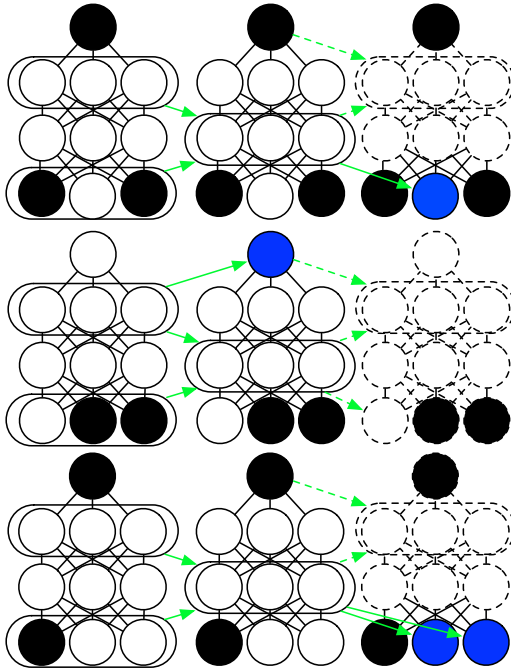

Figure 2: *Multi-prediction training*: This diagram shows the neural nets instantiated to do multi-prediction training on one minibatch of data. The three rows show three different examples. Black circles represent variables the net is allowed to oberve. Blue circles represent prediction targets. Green arrows represent computational dependencies. Each column shows a single mean field fixed point update. Each mean field iteration consists of two fixed point updates. Here we show only one iteration to save space, but in a real application MP training should be run with 5-15 iterations.

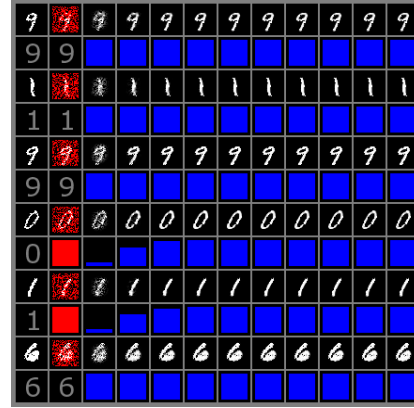

Figure 3: *Mean field inference applied to MNIST digits.* Within each pair of rows, the upper row shows pixels and the lower row shows class labels. The first column shows a complete, labeled example. The second column shows information to be masked out, using red pixels to indicate information that is removed. The subsequent columns show steps of mean field. The images show the pixels being filled back in by the mean field inference, and the blue bars show the probability of the correct class under the mean field posterior.

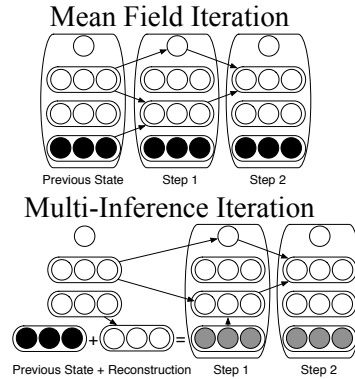

Figure 4: *Multi-inference trick:* When estimating $y$ given $v$, a mean field iteration consists of first applying a mean field update to $h^{(1)}$ and $y$, then applying one to $h^{(2)}$. To use the multi-inference trick, start the iteration by computing $r$ as the mean field update $v$ would receive if it were not observed. Then use $0.5(r + v)$ in place of $v$ and run a regular mean field iteration.

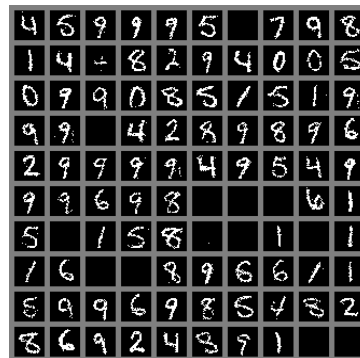

Figure 5: Samples generated by alternately sampling $S_i$ uniformly and sampling $\mathcal{O}_{-S_i}$ from $Q_i(\mathcal{O}_{-S_i})$.

This training procedure is similar to one introduced by Brakel *et al.* [6] for time-series models. The primary difference is that we use $\log Q$ as the loss function, while Brakel *et al.* [6] apply hard-coded loss functions such as mean squared error to the predictions of the missing values.

## 4.2 The Multi-Inference Trick

Mean field inference can be expensive due to needing to run the fixed point equations several times in order to reach convergence. In order to reduce this computational expense, it is possible to train using fewer mean field iterations than required to reach convergence. In this case, we are no longer necessarily minimizing $J$ as written, but rather doing partial training of a large number of fixed-iteration recurrent nets that solve related problems.

We can approximately take the geometric mean over all predicted distributions $Q$ (for different subsets $S_i$) and renormalize in order to combine the predictions of all of these recurrent nets. This way, imperfections in the training procedure are averaged out, and we are able to solve inference tasks even if the corresponding recurrent net was never sampled during MP training.

In order to approximate this average efficiently, we simply take the geometric mean at each step of inference, instead of attempting to take the correct geometric mean of the entire inference process. See Fig. 4 for a graphical depiction of the method. This is the same type of approximation used to take the average over several MLP predictions when using dropout [10]. Here, the averaging rule is slightly different. In dropout, the different MLPs we average over either include or exclude each variable. To take the geometric mean over a unit $h_j$ that receives input from $v_i$, we average together the contribution $v_i W_{ij}$ from the model that contains $v_i$ and the contribution 0 from the model that does not. The final contribution from $v_i$ is $0.5 v_i W_{ij}$ so the dropout model averaging rule is to run an MLP with the weights divided by 2.

For the multi-inference trick, each recurrent net we average over solves a different inference problem. In half of the problems, $v_i$ is observed, and contributes $v_i W_{ij}$ to $h_j$'s total input. In the other half of the problems, $v_i$ is inferred. In contrast to dropout, $v_i$ is never completely absent. If we represent the mean field estimate of $v_i$ with $r_i$, then in this case that unit contributes $r_i W_{ij}$ to $h_j$'s total input. To run multi-inference, we thus replace references to $v$ with $0.5(v + r)$, where $r$ is updated at each mean field iteration. The main benefit to this approach is that it gives a good way to incorporate information from many recurrent nets trained in slightly different ways. If the recurrent net corresponding to the desired inference task is somewhat suboptimal due to not having been sampled enough during training, its defects can be oftened be remedied by averaging its predictions with those of other similar recurrent nets. The multi-inference trick can also be understood as including an input denoising step built into the inference. In practice, multi-inference mostly seems to be beneficial if the network was trained without letting mean field run to convergence. When the model was trained with converged mean field, each recurrent net is just solving an optimization problem in a graphical model, and it doesn't matter whether every recurrent net has been individually trained. The multi-inference trick is mostly useful as a cheap alternative when getting the absolute best possible test set accuracy is not as important as fast training and evaluation.

## 4.3 Justification and advantages

In the case where we run the recurrent net for predicting $Q$ to convergence, the multi-prediction training algorithm follows the gradient of the objective function $J$. This can be viewed as a mean field approximation to the generalized pseudolikelihood.

While both pseudolikelihood and likelihood are asymptotically consistent estimators, their behavior in the limited data case is different. Maximum likelihood should be better if the overall goal is to draw realistic samples from the model, but generalized pseudolikelihood can often be better for training a model to answer queries conditioning on sets similar to the $S_i$ used during training.

Note that our variational approximation is not quite the same as the way variational approximations are usually applied. We use variational inference to ensure that the distributions we shape using backprop are as close as possible to the true conditionals. This is different from the usual approach to variational learning, where $Q$ is used to define a lower bound on the log likelihood and variational inference is used to make the bound as tight as possible.

In the case where the recurrent net is not trained to convergence, there is an alternate way to justify MP training. Rather than doing variational learning on a single probabilistic model, the MP procedure trains a family of recurrent nets to solve related prediction problems by running for some fixed number of iterations. Each recurrent net is trained only on a subset of the data (and most recurrent nets are never trained at all, but only work because they share parameters with the others). In this case, the multi-inference trick allows us to justify MP training as approximately training an ensemble of recurrent nets using bagging.

Stoyanov *et al.* [20] have observed that a training strategy similar to MPT (but lacking the multi-inference trick) is useful because it trains the model to work well with the inference approximations it will be evaluated with at test time. We find these properties to be useful as well. The choice of this type of variational learning combined with the underlying generalized pseudolikelihood objective makes an MP-DBM very well suited for solving approximate inference problems but not very well suited for sampling.

Our primary design consideration when developing multi-prediction training was ensuring that the learning rule was state-free. PCD training uses persistent Markov chains to estimate the gradient. These Markov chains are used to approximately sample from the model, and only sample from approximately the right distribution if the model parameters evolve slowly. The MP training rule does not make any reference to earlier training steps, and can be computed with no burn in. This means that the accuracy of the MP gradient is not dependent on properties of the training algorithm such as the learning rate which can easily break PCD for many choices of the hyperparameters.

Another benefit of MP is that it is easy to obtain an unbiased estimate of the MP objective from a small number of samples of $v$ and $i$. This is in contrast to the log likelihood, which requires estimating the log partition function. The best known method for doing so is AIS, which is relatively expensive [16]. Cheap estimates of the objective function enable early stopping based on the MP-objective (though we generally use early stopping based on classification accuracy) and optimization based on line searches (though we do not explore that possibility in this paper).

## 4.4 Regularization

In order to obtain good generalization performance, Salakhutdinov and Hinton [18] regularized both the weights and the activations of the network.

Salakhutdinov and Hinton [18] regularize the weights using an L2 penalty. We find that for joint training, it is critically important to not do this (on the MNIST dataset, we were not able to find any MP-DBM hyperparameter configuration involving weight decay that performs as well as layerwise DBMs, but without weight decay MP-DBMs outperform DBMs). When the second layer weights are not trained well enough for them to be useful for modeling the data, the weight decay term will drive them to become very small, and they will never have an opportunity to recover. It is much better to use constraints on the norms of the columns of the weight vectors as done by Srebro and Shraibman [19].

Salakhutdinov and Hinton [18] regularize the activities of the hidden units with a somewhat complicated sparsity penalty. See `http://www.mit.edu/~rsalakhu/DBM.html` for details. We use $\max(|\mathbb{E}_{h \sim Q(h)}[h] - t| - \lambda, 0)$ and backpropagate this through the entire inference graph. $t$ and $\lambda$ are hyperparameters.

## 4.5 Related work: centering

Montavon and Müller [15] showed that an alternative, "centered" representation of the DBM results in successful generative training without a greedy layerwise pretraining step. However, centered DBMs have never been shown to have good classification performance. We therefore evaluate the classification performance of centering in this work. We consider two methods of variational PCD training. In one, we use Rao-Blackwellization [5, 11, 17] of the negative phase particles to reduce the variance of the negative phase. In the other variant ("centering+"), we use a special negative phase that Salakhutdinov and Hinton [18] found useful. This negative phase uses a small amount of mean field, which reduces the variance further but introduces some bias, and has better symmetry with the positive phase. See `http://www.mit.edu/~rsalakhu/DBM.html` for details.

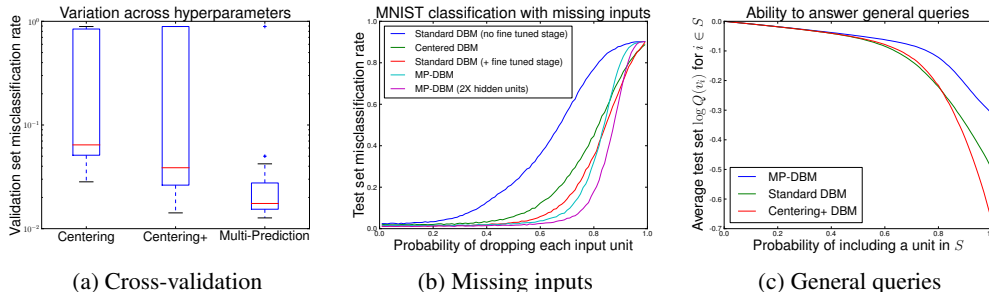

| (a) Cross-validation | (b) Missing inputs | (c) General queries |

Figure 6: Quantitative results on MNIST: (a) During cross-validation, MP training performs well for most hyperparameters, while both centering and centering with the special negative phase do not perform as well and only perform well for a few hyperparameter values. Note that the vertical axis is on a log scale. (b) Generic inference tasks: When classifying with missing inputs, the MP-DBM outperforms the other DBMs for most amounts of missing inputs. (c) When using approximate inference to resolve general queries, the standard DBM, centered DBM, and MP-DBM all perform about the same when asked to predict a small number of variables. For larger queries, the MP-DBM performs the best.

## 4.6 Sampling, and a connection to GSNs

The focus of this paper is solving inference problems, not generating samples, so we do not investigate the sampling properties of MP-DBMs extensively. However, it is interesting to note that an MP-DBM can be viewed as a collection of dependency networks [9] with shared parameters. Dependency networks are a special case of generative stochastic networks or GSNs (Bengio *et al.* [3], section 3.4). This means that the MP-DBM is associated with a distribution arising out of the Markov chain in which at each step one samples an $S_i$ uniformly and then samples $\mathcal{O}$ from $Q_i(\mathcal{O})$. Example samples are shown in figure 5. Furthermore, it means that if MPT is a consistent estimator of the conditional distributions, then MPT is a consistent estimator of the probability distribution defined by the stationary distribution of this Markov chain. Samples drawn by Gibbs sampling in the DBM model do not look as good (probably because the variational approximation is too damaging). This suggests that the perspective of the MP-DBM as a GSN merits further investigation.

# 5 Experiments

## 5.1 MNIST experiments

In order to compare MP training and centering to standard DBM performance, we cross-validated each of the new methods by running 25 training experiments for each of three conditions: centered DBMs, centered DBMs with the special negative phase ("Centering+"), and MP training.

All three conditions visited exactly the same set of 25 hyperparameter values for the momentum schedule, sparsity regularization hyperparameters, weight and bias initialization hyperparameters, weight norm constraint values, and number of mean field iterations. The centered DBMs also required one additional hyperparameter, the number of Gibbs steps to run for variational PCD. We used different values of the learning rate for the different conditions, because the different conditions require different ranges of learning rate to perform well. We use the same size of model, minibatch and negative chain collection as Salakhutdinov and Hinton [18], with 500 hidden units in the first layer, 1,000 hidden units in the second, 100 examples per minibatch, and 100 negative chains. The energy function for this model is

$$E(v, h, y) = -v^T W^{(1)} h^{(1)} - h^{(1)T} W^{(2)} h^{(2)} - h^{(2)T} W^{(3)} y$$
$$-v^T b^{(0)} - h^{(1)T} b^{(1)} - h^{(2)T} b^{(2)} - y^T b^{(3)}.$$

See Fig. 6a for the results of cross-validation. On the validation set, MP training consistently performs better and is much less sensitive to hyperparameters than the other methods. This is likely because the state-free nature of the learning rule makes it perform better with settings of the learning rate and momentum schedule that result in the model distribution changing too fast for a method based on Markov chains to keep up.

When we add an MLP classifier (as shown in Fig. 1d), the best "Centering+" DBM obtains a classification error of 1.22% on the test set. The best MP-DBM obtains a classification error of 0.88%. This compares to 0.95% obtained by Salakhutdinov and Hinton [18].

If instead of adding an MLP to the model, we simply train a larger MP-DBM with twice as many hidden units in each layer, and apply the multi-inference trick, we obtain a classification error rate of 0.91%. In other words, we are able to classify nearly as well using a single large DBM and a generic inference procedure, rather than using a DBM followed by an entirely separate MLP model specialized for classification.

The original DBM was motivated primarily as a generative model with a high AIS score and as a means of initializing a classifier. Here we explore some more uses of the DBM as a generative model. Fig. 6b shows an evaluation of various DBM's ability to classify with missing inputs. Fig. 6c shows an evaluation of their ability to resolve queries about random subsets of variables. In both cases we find that the MP-DBM performs the best for most amounts of missing inputs.

## 5.2 NORB experiments

NORB consists of $96 \times 96$ binocular greyscale images of objects from five different categories, under a variety of pose and lighting conditions. Salakhutdinov and Hinton [18] preprocessed the images by resampling them with bigger pixels near the border of the image, yielding an input vector of size 8,976. We used this preprocessing as well. Salakhutdinov and Hinton [18] then trained an RBM with 4,000 binary hidden units and Gaussian visible units to preprocess the data into an all-binary representation, and trained a DBM with two hidden layers of 4,000 units each on this representation. Since the goal of this work is to provide a single unified model and training algorithm, we do not train a separate Gaussian RBM. Instead we train a single MP-DBM with Gaussian visible units and three hidden layers of 4,000 units each. The energy function for this model is

$$E(v, h, y) = -(v - \mu)^T \beta W^{(1)} h^{(1)} - h^{(1)T} W^{(2)} h^{(2)} - h^{(2)T} W^{(3)} h^{(3)} - h^{(3)T} W^{(4)} y$$
$$+ \frac{1}{2}(v - \mu)^T \beta (v - \mu) - h^{(1)T} b^{(1)} - h^{(2)T} b^{(2)} - h^{(3)T} b^{(3)} - y^T b^{(4)}.$$

where $\mu$ is a learned vector of visible unit means and $\beta$ is a learned diagonal precision matrix.

By adding an MLP on top of the MP-DBM, following the same architecture as Salakhutdinov and Hinton [18], we were able to obtain a test set error of 10.6%. This is a slight improvement over the standard DBM's 10.8%.

On MNIST we were able to outperform the DBM without using the MLP classifier because we were able to train a larger MP-DBM. On NORB, the model size used by Salakhutdinov and Hinton [18] is already as large as we are able to fit on most of our graphics cards, so we were not able to do the same for this dataset. It is possible to do better on NORB using convolution or synthetic transformations of the training data. We did not evaluate the effect of these techniques on the MP-DBM because our present goal is not to obtain state-of-the-art object recognition performance but only to verify that our joint training procedure works as well as the layerwise training procedure for DBMs. There is no public demo code available for the standard DBM on this dataset, and we were not able to reproduce the standard DBM results (layerwise DBM training requires significant experience and intuition). We therefore can't compare the MP-DBM to the original DBM in terms of answering general queries or classification with missing inputs on this dataset.

## 6 Conclusion

This paper has demonstrated that MP training and the multi-inference trick provide a means of training a single model, with a single stage of training, that matches the performance of standard DBMs but still works as a general probabilistic model, capable of handling missing inputs and answering general queries. We have verified that MP training outperforms the standard training procedure at classification on the MNIST and NORB datasets where the original DBM was first applied. We have shown that MP training works well with binary, Gaussian, and softmax units, as well as architectures with either two or three hidden layers. In future work, we hope to apply the MP-DBM to more practical applications, and explore techniques, such as dropout, that could improve its performance further.

### Acknowledgments

We would like to thank the developers of Theano [4, 2], Pylearn2 [7]. We would also like to thank NSERC, Compute Canada, and Calcul Québec for providing computational resources.

## Footnotes

[1]Code and hyperparameters available at `http://www-etud.iro.umontreal.ca/~goodfeli/mp_dbm.html`

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
