[Reviews · NeurIPS 2013]

Submitted by Assigned_Reviewer_5

The paper presents a method for learning layers of representation and for completing missing queries both in input and labels in single procedure unlike some other methods like deep boltzmann machines (DBM). It is a recurrent net following the same operations as DBM with the goal of predicting a subset of inputs from its complement. Parts of paper are badly written, especially model explanation and multi-inference section, nevertheless the paper should be published and I hope the authors will rewrite them.

Details:
- The procedure is taken from DBM, however other then that, is there a relation between the DBM and this algorithm, or should we just treat the algorithm as one particular function (recurrent net (RNN)) that predicts subset of inputs from its complement? If there is no relation may be the algorithm should have a different name. Or is the fact that procedure was taken from DBM the reason that the algorithm works? What if I tried other types of RNN's with the same objectives? The one special thing that is interesting here is that the RNN can be run till convergence.
- If the network is like DBM one should be able to generate from it without any inputs (after training of course). Can you do this? If yes, show the result, if not explain why. If I understand correctly in one of the experiments you only clamp the label and fill the input. The net should produce variety of inputs from the same class. Does it? Can you show them?
- It is not surprising that the network does better then DBM on input completion - it was specifically trained to do so, so it's not a fair comparison, though good experiment nevertheless.
- Section 4: You haven't defined Q and P in the formulas. What are they? Please write formulas of your model (not needed for those of the backpropagation). If the model can be thought of more generally, write the basic version explicitly (e.g. an algorithm would be demonstrative).
- Figure 2: You haven't defined r. I don't understand why do you average your input with with r.
- Figure 2: "Green-boxed rows represent class variabels" - how can they - there are pictures of digits there not classes labels. What are these?

Quality: Good
Clarity: Bad
Originality: While most of the separate ideas of this paper are known in one way or another, they take a reasonable form here and overall form a good novel idea.
Significance: Potentially significant.
Summary: The idea is interesting, works and it is potentially important. Parts of the paper are very unclear and should be rewritten.

Submitted by Assigned_Reviewer_6

This paper introduces the MP-DBM as an alternative to DBMs that allow
to avoid DBM greedy training, and outperforms DBM at classification
and classification with missing inputs on two datasets.

In general, the proposed ideas seems valuable and well motivated. Unfortunately,
the paper is unnecessarily difficult to follow. The presentation is
very verbose without much formal definitions of the models and
algorithms. I would suggest the authors to focus on one aspect of the
problem, and take the required space to define all the components
formally, and empirically evaluate them (see specific comments below).

Too much space is devoted to motivation in Section 3. Instead, more
space should be taken for a much more formal description of the model.


In particular, the main algorithm of the paper is described in one
sentence (lines 208-209) and exemplified in one third of a Figure
(2a). A much more detailed description (i.e. application of the mean
field point equation to the actual layers) should be provided. The
readers must be able as much as possible to reproduce the experiments
from the paper.

Similarly, Section 5 should contain equations and formal algorithms to
support the text.

Section 7: Please support your claims ("for joint training, it is
critically important not to [...] regularize the weights using a L2
penalty"). Also, please discuss your choice of regularization (line
303).

Section 9: Define the hyperparameters, and the learning algorithm.

Line 355: What is the "fine-tuning" you mention? Please define.

Figure 3: How does the reported error rates on MNIST compare to the
state-of-the-art?

EDIT: I have taken into account authors clarifications in the rebuttal and updated
the score. That being said, most of my comments about clarity stated above remain valid:
I expect the authors to add formal descriptions of their model and learning algorithm
in the camera-ready version, as they promise in their rebuttal.

The relationship between this model and dropout could be further
discussed.

Minor comments:

The captions in Fig. 2 are unusually long. It would be more elegant to
put the details in the text itself.

Line 198: y is absent from the equation, I assume it is simply
considered as an additional dimension in v?

Section 8: Please define the terminology. E.g. what is the condition
number? The Hessian?

Section 9: Please minimally describe the MNIST dataset.
Summary: The MP-DPM presented in this paper is well motivated, and seems to solve several issues arising with the standard DBMs. Unfortunately, the presentation of the paper is very verbose and lack focus, so it is difficult for the reader to assess the validity of the approach, as well as the significance of the reported empirical results.

Submitted by Assigned_Reviewer_7

Deep Boltzmann Machines (DBNs) are usually initialized by greedily training a stack of RBMs, and then fine-tuning the overall model using persistent contrastive divergence (PCD). To perform classification, one typically provides the mean-field features to a separate classifier (e.g. a MLP) which is trained discriminatively. Therefore the overall process is somewhat ad-hoc, consisting of L + 2 models (where L is the number of hidden layers) each with its own objective. This paper presents a holistic training procedure for DBNs which has a single training stage (where both input and output variables are predicted) producing models which can classify directly as well as efficiently performing other tasks such as imputing missing inputs. The main technical contribution is the mechanism by which training is performed; a way of training DBNs which uses the mean field equations for the DBN to induce recurrent nets that are trained to solve different inference tasks (essentially predicting different subsets of observed variables).

Quality: I think this is a really interesting paper and its implications to DBN training are important. I agree with the authors that, despite DBNs being really powerful models, the generally accepted way of training them has several shortcomings. The authors explicitly point these out in section 3: greedily training is sub-optimal, using multiple task-specific models loses some benefit of probabilistic modeling, and the greedy training procedure is overly complex. The motivation is strong, the methods are presented clearly, the work is intuitive and clever, and the experiments demonstrate that the method works (even if the only datasets considered are the old MNIST and small NORB).

Clarity: Generally the paper is well written. There are far too many top-level sections (11)! Combine 4- 8 into Methods, and 9-10 into Experiments. Also, even after reading the caption for figure 2 c) several times, I still didn't fully understand it.

Originality: To the best of my knowledge, the approach is original (an earlier version appeared at ICLR workshop track).

Significance: The paper is important to the deep learning/representation learning community. In fact, by simplifying the DBN training procedure, the paper may encourage more DBN activity specifically those who don't have experience with layer-wise training procedures (as the authors point out in section 1). I wouldn't be surprised if this work doesn't lead to more subset-like training procedures for other architectures. The multi-inference "trick" described in section 5 is interesting in its own right (separate from the DBN application).

Comments

- Even though "y" is almost universally used for output and "x" input, the second paragraph in section refers to these variables without defining them. Please define these variables before using them.

- With regards to not being able to train a larger MP-DBM on NORB (and thus validate the effectiveness of holistic prediction on more than 1 dataset), there is no reason not to try distributing the model over multiple GPUs. As well, recent work "Predicting Parameters in Deep Learning" (Denil et al. Arxiv) gives another way to reduce effective model size but retain capacity without having to go to distributed models.

Summary: A novel, effective, simpler way to train Deep Boltzmann Machines. More extensive experiments could further justify the method.
Author Feedback

Author rebuttal: It sounds like the main concerns with this paper are related to clarity. We will certainly rewrite the paper based on your feedback. In the response we try to clarify as much as possible so you can write a more confident review.


Where to find definitions:

Q: line 198 and 207. We will add more explanation. Q is defined implicitly by solving an optimization problem, so there is not an explicit formula for it.

P (R5): line 73. We will add the specific energy functions for MNIST and NORB to their respective experimental sections.

r (R5): line 234

y (R6): line 66


R5
-To what extent is this still a DBM?

It still is a DBM. You can gain some understanding by also thinking of it as a family of recurrent nets. Both views are valid.

-Would the RNN still work if it weren't taken from the DBM?

Not trivially. The RNN is just mean field in the DBM. Without the DBM and mean field theory, we wouldn't know which recurrent net to run to solve which inference problem. There are probably very many other ways of mapping inference problems to inference nets, but that's a broad research topic this paper does not attempt to address.

-Can it generate samples?

Yes, but not as high quality of samples as the DBM because the training algorithm deprioritizes that relative to inference. See lines 256-260. We can add pictures of samples.

-If I understand correctly in one of the experiments you only clamp the label and fill the input. The net should produce variety of inputs from the same class.

No, we didn't do that in this paper. If you're thinking of figure 2c, we are clamping the input and filling in the label.

The MP-DBM model is still a DBM, so it is still capable of filling in multiple solutions, regardless of what you clamp. But to do this you need to run Gibbs sampling, not mean field. We've only shown results of mean field in this paper.

- It's not fair to compare the MP-DBM and DBM on input completion.

Please note that the MP-DBM is also better at classifying with missing examples. The DBM is trained in a generative fashion so it should be able to do this if it is really able to represent a probability distribution well.

- Figure 2c: "Green-boxed rows represent class variabels" - how can they - there are pictures of digits there not classes labels. What are these?

The digit displayed in the leftmost box indicates the true class (we use images of the class to represent the class to the reader, since the images are human-readable). The second box shows the input to the model's label field. If it shows the digit again, the model is given the label as input. If there is a red box there, the model is asked to fill in the label. Subsequent boxes show averages images of the 10 digits, weighted by the model's posterior.


R6

We will rewrite to fit more description of the model.

We will move the text out of the captions for figure 2 and into the main text.

Reproducibility of experiments, hyperparameters: See footnote 1. We will release all code, so experiments will be perfectly reproducible. All hyperparameters are included in the supplementary material.

Section 5 does contain a formal description of the method, it's just written as prose with variables rather than equations. We will add a version formatted as equations.

"Section 7: Please support your claims ("for joint training, it is critically important not to [...] regularize the weights using a L2 penalty")."

Our best classification result on MNIST without weight decay is 0.88% test error. Our best result with weight decay was 1.19%. We obtained the 0.88% after running less than 50 training jobs, but the 1.19% took months of effort and thousands of training jobs to obtain.

"please discuss your choice of regularization"

It is similar to the regularization that S&H 2009, but simpler. Their regularization doesn't correspond to a simple term added to the loss function. They apply a penalty to the hidden unit activations but then during backprop, they backprop the error derivatives through a feedforward linear network rather than the sigmoidal recurrent network that actually produced the hidden units. We decided to add a function of the true inference graph, since one of our main goals is conceptual simplicity.

Section 9: Define the learning algorithm.

It's defined in section 4.

Terminology:

fine-tuning: The addition of the MLP used by Salakhutdinov and Hinton, as shown in figure 1c, trained with respect to a (supervised) classification training criterion.

Hessian, condition number: we will add a link to a tutorial, such as http://www.math.cmu.edu/~shlomo/VKI-Lectures/lecture1/node5.html

What is the state of the art on MNIST?

Without using any image-specific prior (as in this paper) it is 0.79% error, using dropout. We hope to explore dropout enhancement of MP-DBMs in a later paper. Our goal for this paper was just to replace the layer wise DBM training algorithm with a joint training algorithm.

-Line 198: is y included in v?

Yes.



R7

We will reduce the number of top-level sections as you suggest.

Fig 2c: We obviously need to work on this one; we'll run it by some people who weren't involved with the paper before the final copy. Does our clarification to R5 help at all?

Multi-machine scaling:

We're definitely interested in that, but it's a non-trivial engineering challenge.

Misha Denil's work:

This is certainly interesting! I think the amount it can reduce the memory requirements on a single machine is limited though, because the parameters must be decompressed for the learning algorithm to work. The main benefit of that work is that it reduces the number of parameters that must be communicated between machines before they are decompressed locally.

On scaling in general:

Here we constrained ourselves to densely connected models, in order to compare against the existing DBM work. Convolutional models should scale much better.